# Gut Microbiota and Breast Cancer: The Dual Role of Microbes

**DOI:** 10.3390/cancers15020443

**Published:** 2023-01-10

**Authors:** Ana Isabel Álvarez-Mercado, Ana del Valle Cano, Mariana F. Fernández, Luis Fontana

**Affiliations:** 1Department of Biochemistry and Molecular Biology 2, School of Pharmacy, Campus de Cartuja s/n, 18071 Granada, Spain; 2Institute of Nutrition and Food Technology “José Mataix”, Biomedical Research Center, Parque Tecnológico Ciencias de la Salud, Avda. del Conocimiento s/n, Armilla, 18016 Granada, Spain; 3Instituto de Investigación Biosanitaria ibs.GRANADA, Complejo Hospitalario Universitario de Granada, 18071 Granada, Spain; 4Department of Radiology, School of Medicine, and Biomedical Research Center, University of Granada, 18071 Granada, Spain; 5Spanish Consortium for Research on Epidemiology and Public Health (CIBERESP), 28029 Madrid, Spain

**Keywords:** androbolome, breast cancer, estrobolome, microbiome, microbiota

## Abstract

**Simple Summary:**

The set of microbes in our body, called microbiota, exerts a wide variety of beneficial effects and is related to the state of health of the person. An alteration in the composition of the microbiota is called dysbiosis and is related to the state of the disease. Microbiota exists in many locations in our body, but the most important from a quantitative point of view is the intestinal microbiota, which is why it is the most studied. However, our microbiota is also capable of producing harmful effects, thereby in recent years it has been considered another environmental factor to be taken into account in the risk of developing diseases, including cancer.

**Abstract:**

Breast cancer is the most frequently diagnosed cancer and also one of the leading causes of mortality among women. The genetic and environmental factors known to date do not fully explain the risk of developing this disease. In recent years, numerous studies have highlighted the dual role of the gut microbiota in the preservation of host health and in the development of different pathologies, cancer among them. Our gut microbiota is capable of producing metabolites that protect host homeostasis but can also produce molecules with deleterious effects, which, in turn, may trigger inflammation and carcinogenesis, and even affect immunotherapy. The purpose of this review is to describe the mechanisms by which the gut microbiota may cause cancer in general, and breast cancer in particular, and to compile clinical trials that address alterations or changes in the microbiota of women with breast cancer.

## 1. Introduction

In recent years, numerous studies have highlighted the dual role of the gut microbiota in the preservation of host health and in the development of different pathologies [1], cancer being one of the most studied. The microbes that inhabit our gut are capable of producing a series of metabolites that protect host homeostasis but, in situations of dysbiosis, they can also produce molecules with deleterious effects which, in turn, may trigger inflammation and carcinogenesis [2].

Cancer is a multifactorial disease that represents the second leading cause of death worldwide, accounting for nearly 10 million deaths in 2020 [3,4]. The incidence of breast cancer (BC) has risen worldwide to unprecedented levels in recent decades, making it the major cancer of women in many parts of the world nowadays. It is not only the most frequently diagnosed cancer (excluding non-melanoma skin cancers) among women, affecting one in eight women during their lifetime, but also one of the leading causes of cancer mortality in women, with 684,996 deaths in 2020 [3,4].

Genetic and environmental factors do not fully explain each person’s risk of developing the disease, since some individuals, sometimes genetically identical, who have similar lifestyles and ages, develop cancer, while others do not. The random occurrence of replicative errors in DNA that results in different types of mutations is part of the explanation, and this DNA alteration process appears to be related to the composition and function of the microbiota. The cancer–microbiota relationship has been found both in local gastrointestinal cancers and in other types of tumors. Nowadays, it is clear that certain infectious pathogens, such as *Helicobacter pylori*, human papillomavirus, and hepatitis B and C viruses are strong causes of cancer [5]. However, these are just a few, so much so, that in 2017, the International Consortium Cancer Microbiome (ICMC, https://www.icmconsortium.org/ (accessed on 16 July 2022) was founded in recognition of the emerging importance of the human microbiome in oncology. The ICMC is made up of clinical experts in oncology and the microbiome, its aim being to promote microbiome research within the field of oncology, establish expert consensus, and deliver education for academics and clinicians [6].

However, metabolomic and metagenomic studies have revealed that the gut microbiota not only affects carcinogenesis itself, but also cancer prevention and therapy, and that microbes may act through various mechanisms sometimes opposite to each other (e.g., microorganisms are capable to act as tumor suppressors or, conversely, as oncogenic), giving rise to a complex and bidirectional relationship. The purpose of this review is to describe the mechanisms by which the gut microbiota may cause cancer in general, and BC in particular, and to compile clinical trials that address alterations or changes in the microbiota of women with BC.

## 2. Beneficial Effects Exerted by the Gut Microbiota

The intestinal microbiota is beneficial because it exerts a wide variety of positive health effects through different mechanisms which have been extensively reviewed [7,8]. First, it reinforces the intestinal barrier by stimulating mucus production by intestinal epithelial cells (IECs), strengthening the tight junctions that are also established between IECs, and stimulating the secretion of immunoglobulin A (sIgA) by immune cells present in the intestine.

Secondly, the components of the microbiota can compete with pathogenic microbes for binding to the intestinal mucosa (competitive exclusion), or directly prevent/inhibit the binding of pathogens to the intestinal mucosa.

Thirdly, the microbiota produces a wide range of molecules with a variety of biological activities: short-chain fatty acids (SCFA), acetate, propionate, and butyrate, which serve as energy substrates for the IECs; vitamins, such as K, cobalamin, biotin, and folic acid, among others; hormones, such as catecholamines; and neurotransmitters, such as acetylcholine, serotonin, and dopamine. Many of these molecules can be absorbed and distributed to other organs, hence the existence of a brain–gut or liver–gut axis. In the case of the nervous system, the neurotransmitters produced by the microbiota can affect it through their influence on the neurons that innervate the intestine, which, via the vagal pathway, reach the brain. Other molecules that can be included in this section are peptides with antimicrobial activity, referred to as bacteriocins, such as bifidocin A and lactacin, which act by inhibiting the synthesis of the bacterial wall or by inducing the formation of pores in this wall, and compounds with antifungal activity, such as benzoic acid.

Finally, the microbiota exerts immunomodulatory effects, which take place, among others, thanks to the interaction with antigen-presenting cells, such as dendritic cells, which emit very long projections capable of reaching the intestinal lumen, and by interacting with the Toll-like receptor (TLR) cascade signaling. In this sense, bacterial lipopolysaccharide (LPS), a major component of the outer membrane in Gram-negative bacteria, may activate the host’s cell surface Toll-like receptor 4 (TLR4), thus triggering immune T cell-mediated response against cancer cells [9].

## 3. Detrimental Effects Exerted by the Gut Microbiota and Their Relationship with Cancer

In a recent work, Nejman et al. (2020) [10] studied the human tumor microbiome. These authors investigated seven types of solid cancers (breast, lung, ovarian, pancreatic, bone, skin, and brain) and obtained very interesting results that can be summarized as follows: (1) bacterial components, such as DNA, RNA, and lipopolysaccharide (LPS, a component of the wall of Gram-negative bacteria), were detected in the seven tumors. Lipoteichoic acid, a component of the wall of Gram-positive bacteria, was also detected, but only in skin cancer, and, to a much lesser extent, in BC. (2) That human tumors contain bacteria has been known for a long time. However, another novelty of this work is that tumor bacteria are located inside cancer cells. In fact, both Gram-positive and Gram-negative bacteria were detected inside tumor cells and immune cells, such as macrophages and CD45+ leukocytes. Bacteria were always found in the cytosol near the nucleus but never inside the nucleus and, moreover, they lacked their cell wall. (3) Each tumor exhibited a different microbiota, but that of BC was the richest and most diverse compared to that of other types of cancer. The authors even provided some metabolomic insights. In the case of BC, characterized by high oxidative stress, an abundance of mycothiol-producing bacteria, an agent involved in the elimination of reactive oxygen species, was found.

However, more important as a cancer risk and promoting factor than the microbiota composition is its functionality. Several possible mechanisms have been proposed to understand the microbial influence on cancer (Figure 1).

### 3.1. Degradation of p53

The first pathogenic bacterium involved in the development of cancer was *Helicobacter pylori*, classified as a class I carcinogen by the World Health Organization (WHO). This bacterium produces the virulence factor CagA (the product of the cytotoxin-associated gene A), which induces the degradation of the p53 tumor suppressor gene in gastric epithelial cells, thus promoting the increase in gastric cancer [11]. Another example is *Shigella flexneri*, which interferes with DNA damage response and repair pathways, also inducing host’s cell degradation of p53, through the secretion of its enzymes inositol-phosphate phosphatase D (IpgD) and cysteine protease-like virulence gene A (VirA), thus increasing the probability of occurrence of mutations during the repair response of damaged DNA of infected cells [12].

### 3.2. Genomic Instability and DNA Damage

Although DNA damage may not be sufficient in itself to promote cancer development, double-strand breaks are the most detrimental type of DNA damage caused by genotoxins, reactive oxygen species, and ionizing radiation [13]. Urbaniak et al. (2016) [14] examined the ability to induce DNA double-stranded breaks of bacteria isolates cultured from normal adjacent tissue of BC patients. They found that several *Escherichia coli* isolates and one *Staphylococcus epidermidis* isolate displayed this ability through the production of colibactin, which could cause genomic instability. *Bacillus*, *Micrococcus,* and *Propionibacterium* isolates did not induce double-strand breaks [14]. Other members of the *Enterobacteriaceae* family can also produce colibactin [6].

Another toxin with DNAse activity produced by Gram-negative bacteria is the cytolethal distending toxin (CDT). This toxin generates double strand breaks in the DNA of epithelial cells when released in the vicinity of the gastrointestinal epithelium, thus promoting a transient cell cycle arrest and allowing the appearance of mutations that can lead to tumor formation [15]. *E. coli* and *Campylobacter jejuni*, among others, produce CDT [6].

Pathogenic bacteria can indirectly favor carcinogenesis through the generation of oxidative stress. For example, toxins produced by *Bacteroides fragilis* and *Helicobacter pylori* are capable of activating the human enzyme spermine oxidase, which generates hydrogen peroxide and other reactive oxygen species (ROS) capable of causing DNA damage [2,16]. Apart from these toxins, other microorganisms, such as *Enterococcus faecalis*, *Porphyromonas* sp, *Bilophila,* and *Fusobacterium,* are capable of producing extracellular oxygen-derived species and hydrogen sulphide that can penetrate human cells, increasing the oxidative environment and causing DNA mutations [2,6,15].

Finally, pyridoxine (vitamin B_6_) is one of the B vitamins synthesized by bacteria in our microbiota. Pyridoxine deficiency has been shown to decrease serine hydroxymethyltransferase and betaine-homocysteine methyltransferase activities, which reduce the pool of methylene groups for 5,10-methylene-tetrahydrofolic acid, resulting in an increase in the frequency of uracil incorporation during DNA synthesis that may be associated with mutation and DNA strand breaks [17].

### 3.3. Metabolism of Endogenous and Exogenous Compounds

The estrobolome and androbolome are mechanisms that may explain the relationship between the microbiota and hormone-dependent cancers, such as breast and prostate cancers. The estrobolome is the collection of microbial genes responsible for the synthesis of machinery related to estrogen metabolism and, therefore, to its circulating levels [18]. This machinery includes glucuronidases, glucosidases, and dehydrogenases. The androbolome is the equivalent of the estrobolome, but applied to androgen metabolism [19]. Perturbations in the microbiota/estrobolome can, therefore, lead to elevated levels of circulating estrogens and its metabolites, thereby increasing the risk of BC.

The metabolism of estrogens takes places in the liver, where they are conjugated and excreted into the gastrointestinal lumen within the bile; there, they are de-conjugated by bacterial β-glucuronidase, and then they are re-absorbed as free estrogens through enterohepatic circulation, getting to different organs such as the breast [20]. In addition, estrogen-like metabolites can be also produced by oxidative and reductive reactions in the gut and by an induced synthesis of estrogen-inducible growth factors, which might have a carcinogenic potential. Moreover, bacterial β-glucuronidase could participate in the deconjugation of xenobiotics and/or xenoestrogens, leading to their reuptake through the enterohepatic pathway and thus increasing the time they remain in the organism [21]. Many β-glucuronidase bacteria are found in two dominant subgroups, namely, the *Clostridium leptum* cluster and the *Clostridium coccoides* cluster, which belong to the *Firmicutes* phylum. The *Escherichia/Shigella* bacterial group, a member of the *Proteobacteria* phylum, also possesses β-glucuronidase enzymes [22].

β-Glucuronidase could also play a major role in the deconjugation of endocrine disrupting chemicals, such as bisphenol-A, increasing the time that they remain in the organism. Some endocrine compounds could induce alterations in the gut microbiota and the metabolites they produce, which may be associated with increased inflammation [23].

### 3.4. Alteration of Cell Proliferation and Survival Pathways (β-Catenin, MAPK and AKT)

Certain intestinal bacteria can modulate different cell proliferation and survival pathways, thus contributing to cancer. This is the case of the β-catenin pathway. Alterations in this pathway lead to dysregulation of cell growth, acquisition of stem cell-like characteristics, and loss of cell polarity. Different toxins, such as the CagA protein from *Helicobacter pylori*, the FaDa adhesion factor from *Fusobacterium nucleatum,* and the metalloproteinase (MP) toxin from *Bacteroides fragilis* are able to interact, directly or indirectly, with the host’s epithelial cell adhesion molecule E-cadherin, disrupting intercellular junctions and activating β-catenin signaling. This, in turn, triggers cell proliferation and the potential carcinogenic transformation of the affected host’s cells [24,25,26]. Similarly, virulence factor A (AvrA) from *Salmonella enterica* is able totranslocate into host’s cells and activate β-catenin signaling through its deubiquitinase activity [2,27].

Other virulence factors released in the gut during pathogenic infection can induce transformation to cancer cells by infecting pre-transformed cells through activation of other cell survival pathways, such as the mitogen-activated protein kinase kinase (MAPK) and protein kinase B (AKT) pathways. The CagA protein from *Helicobacter pylori* acts on the MAPK pathway and the AvrA factor from *Salmonella enterica* acts by promoting both MAPK and AKT pathways [2,28,29].

### 3.5. Activation of Proinflammatory Pathways

Inflammation is a central feature of carcinogenesis regardless of the etiologic agent and is thought to be the main oncogenic mechanism of the microbiota [30]. Microbial virulence factors induce chronic inflammation of host tissue, stimulating cell proliferation that can ultimately become dysregulated and, when combined with a failure of apoptosis, result in initiation of the carcinogenesis process [6,31].

The loss of integrity of mucosal barriers stimulates pro-inflammatory programs with activation of pathways (such as NF-κB and STAT3) that are known to be involved in carcinogenesis [32]. Thus, *Fusobacterium nucleatum*, associated with colorectal cancer, can induce activation of the nuclear factor-κB (NF-κB) pathway [33] and *Bacteroides fragilis* secretes the aforementioned toxin that stimulates a T helper type-17-dependent colitis and promotes tumorigenesis [34].

There is also evidence that certain microorganisms can induce proinflammatory effects in remote organs through interactions with host’s recognition receptors, such as TLRs and nucleotide-binding oligomerization domain (NOD)-like receptors [6,15]. Interaction between LPS and TLR4 results in the downstream activation of cell survival pathways and has been cited as a mechanism by which the intestinal microbiome may contribute to carcinogenesis outside the gastrointestinal tract [35].

### 3.6. Dysregulation of the Immune System

The immune system plays a key role in preventing carcinogenesis by inducing death in an abnormal host’s cells with neoplastic potential [6]. Although the human microbiota collaborates with the immune system in its anticancer fight through mechanisms such as T-cell receptor amplification and by enhancing the immune response itself, some bacteria may suppress a host’s immunity, thus helping the tumor to be unrecognized by our immune system [15]. Some bacteria can stimulate carcinogenesis by blocking immune mechanisms that normally keep it inhibited. For example, *Fusobacterium nucleatum* inhibits T cells and NK cells, through the bacterial virulence factor Fap2, able to bind and block the NK inhibitory TIGIT factor, thus stopping the NK attack against tumor cells [36].

### 3.7. Epigenetic Mechanisms

Epigenetics encompasses three distinct, although closely related, mechanisms that regulate gene expression without changing the nucleotide sequence: DNA methylation, histone modification, and non-coding RNA. Certain metabolites produced by the microbiota have been described to modulate gene expression epigenetically. Perhaps the most surprising of the metabolites with epigenetic effect is butyrate. This SCFA is an inhibitor of enzymes with histone deacetylase (HDAC) activity [37,38] and, therefore, is capable of activating silenced genes. Thus, butyrate has been shown to derepress genes, such as the cell-cycle inhibitor p21 and the proapoptotic protein Bcl-2 homologous antagonist/killer (BAK) in cancer cells, and to activate these genes in normal cells [37,39], among many others. Other SCFAs, such as acetate, lack this HDAC inhibitory activity [40].

The epigenetic role of other metabolites/compounds produced by the microbiota is not so striking. Folate is a vitamin that participates in the transfer of one-carbon units (methyl, formyl, methenyl, etc), and biotin participates in carboxylation and biotinylation reactions, both of which may affect histone remodeling.

The gut microbiota also contributes to the absorption and excretion of minerals, such as zinc, iodine, selenium, cobalt, and others, that are cofactors of enzymes participating in epigenetic processes. Moreover, various enzymes, such as methyltransferases, acetyltransferases, deacetylases, phosphotransferases, kinases, and synthetases, are derived from the gut microbiota. A number of key metabolites, including the methyl-donor *S*-adenosylmethionine, the acetyl-donor acetyl-CoA, NAD^+^, α-ketoglutarate, and ATP, serve as essential cofactors for many epigenetic enzymes that regulate DNA methylation, posttranslational histone modifications, and nucleosome position [38].

Finally, it is worth noting that the mechanisms described in this section are intertwined and are not sealed compartments. Thus, the aforementioned HDAC inhibitory effect of butyrate promotes IL-12 expression and influences cytotoxic CD8^+^ T cell function, suggesting that manipulation of the gut microbiota could be effective as a part of cancer therapy [41].

## 4. Effects of Microbiota on Clinical Outcomes and Chemotherapy Resistance

### 4.1. Importance of Gut Microbiota in Cancer Therapies

The common goal of the different cancer therapies is to effectively eliminate cancer cells in order to eradicate the disease in the patient and prevent a future recurrence. Despite the great advances in cancer treatments, almost all are also toxic for non-cancerous cells, which leads to the appearance of different side effects of varying severity, some of them even affecting the survival of patients. Gut microbiota and cancer therapies are closely related [2]. Treatments, such as radiotherapy, chemotherapy, and immunotherapy, can modify the microbiota of patients and, at the same time, the composition of the microbiota can influence efficacy and development of side effects of such therapies [42].

As we have seen in previous sections, the gut microbiota can modulate the progression of cancer pathogenesis through its ability to synthesize different antitumor compounds, as well as to regulate the immune response and host inflammatory pathways. These combined mechanisms may explain the strong influence of the microbiota with the efficacy of different therapies.

### 4.2. Intestinal Microbiota and Chemotherapy

The gut microbiota can modulate the metabolism of different drugs used in chemotherapy, thus affecting both the response of cancer cells to this treatment and the susceptibility of healthy cells.

#### 4.2.1. Gemcitabine

Gemcitabine (2′-2′-difluoro-deoxycytidine) is a pyrimidine antagonist, which therefore competes with deoxycytidine (a component of deoxyribonucleic acids derived from cytosine) during DNA synthesis. The antitumor activity of gemcitabine, used in the treatment of different types of cancer, is based on its intracellular activation and subsequent degradation, through its transformation into the inactive metabolite difluoro-deoxy-uridine by cytidine deaminase (CDD) [15]. Studies in mice have concluded that gemcitabine resistance may be due to enhanced metabolic degradation of the drug into difluoro-deoxy-uridine due to the expression of a long isoform of the bacterial enzyme cytidine deaminase (CDDL), which is mainly observed in Gammaproteobacteria [43] On the other hand, the combined action of the antibiotic ciprofloxacin, together with gemcitabine, seems to increase the antitumor activity of the drug through the inhibition of bacterial growth caused by the antibiotic, demonstrating that modulation of the intestinal microbiota can influence the activity of gemcitabine in mice [44].

#### 4.2.2. Cyclophosphamide

Cyclophosphamide is an alkylating agent used in different types of cancer, which acts by stimulating the immune response against cancer. Studies in mice have shown that when cyclophosphamide is administered together with gram-positive bacteria antibiotics, there is an inhibition of the immune response elicited by cyclophosphamide, and therefore of the anticancer effect of the drug, which is restored by oral administration of Gram-positive bacteria, such as *Lactobacillus johonsoni* and *Enterobacter Hirae* [45,46].

#### 4.2.3. Irinotecan

Irinotecan (CPT-11) is an inhibitor of DNA replication through its anti-topoisomerase I action. This drug, used in different types of cancer, has an active form (SN-38) and an inactive form (SN-38-G) that are excreted into the intestine. When SN-38G is excreted into the intestinal lumen, it is converted back to SN-38 by the bacterial ß-glucuronidase of *E. coli*, a process that can cause enteric injury and, therefore, diarrhea, this being one of the main side effects of the drug. In mice, it has been shown that administration of this drug with a bacterial ß-glucuronidase inhibitor can prevent gastrointestinal toxicity [47].

#### 4.2.4. Cisplatin

Cisplatin is an effective anticancer agent and is used in many advanced cancers. It has antibiotic effects on Gram-negative and Gram-positive bacteria and can cause intestinal dysbiosis [48,49]. In addition, cisplatin can also cause loss of intestinal mucosal integrity by binding to the DNA of epithelial cells, impairing their replication, which could lead to serious infections of different parasites [50]. Cisplatin also has other side effects in which the microbiota is involved, such as ototoxicity, mucositis, and weight loss. It has been determined that the administration of D-methionine, together with cisplatin treatment, protects against drug toxicity through, not only its antioxidant and anti-inflammatory properties, but also by promoting the growth of beneficial bacteria, such as *Lachnospiraceae* and *Lactobacillus*, thus regulating the imbalance of the intestinal microbiota [51]. On the other hand, the intestinal microbiota also seems to affect the efficacy of cisplatin. In mice with lung tumors, it has been shown that, when administering this drug with anti-Gram positive antibiotics, the efficacy of the treatment is reduced, as mice survive less and develop larger tumors than mice in which cisplatin is combined with probiotics, such as *Lactobacillus* [49].

#### 4.2.5. 5-fluorouracil

5-fluorouracil (5-FU) is a thymidylate synthase inhibitor used for the treatment of gastrointestinal tumors. Its usefulness is limited due to the acquisition of resistance and the gastrointestinal toxicity effects it causes, one of the most relevant side effects of 5-FU being intestinal mucositis. 5-FU can cause intestinal dysbiosis even with a single dose; different studies have reported a drastic change in the microbiota, decreasing species such as *Bifidobacterium* and *Lactobacillus* and increasing others, such as *Escherichia*, *Clostridium,* and *Enterococcus*. Regarding drug efficacy, it has been shown, in mice, that combined administration with an antibiotic cocktail decreases antitumor efficacy, while probiotic supplementation seems to increase it significantly [52].

Figure 2 summarizes the impact of gut microbiota in several common drugs used in chemotherapy.

### 4.3. Gut Microbiota and Immunotherapy

Immunotherapy is based on immune checkpoint inhibitor (ICI) molecules, which act by blocking certain immune regulatory pathways in order to enhance the antitumor immune response. ICIs are monoclonal antibodies that target receptor molecules on the surface of T lymphocytes, such as cytotoxic lymphocyte antigen 4 (CTLA-4) and programmed death receptor 1 (PD-1), or PD-1 ligands (PD-L1 or PD-L2). The mechanisms of each of these antibodies are different [53].

Because they dysregulate the immune system, ICIs cause a wide spectrum of side effects that can affect any organ. These side effects are known as immune-related adverse events (irAEs), which will differ according to the therapy used. In general, the ICI with the highest incidence and severity of irAEs are antibodies to CTLA-4, followed by those to PD1, with antibodies to PD-L1 having the least effect. In particular, intestinal side effects, such as diarrhea or colitis, are more frequently observed with anti-CTLA-4 antibodies, while dysthyroidism or pulmonary toxicity are more frequent with anti-PD-1/PD-L1 [53]. Because of this, there are a significant number of patients to whom such therapy can be applied only for a limited time due to the occurrence of strong side effects. However, oral administration of certain probiotics, such as *Bacterioides fragilis* and *Burkholderia cepacia*, has been linked to improvement of these immunotherapy-associated side effects [54].

In terms of efficacy, ICIs have demonstrated their usefulness in different solid tumors, as well as in hematologic malignancies. Although ICIs achieve a durable response and prolonged survival, a non-negligible percentage of patients do not obtain any benefit (primary resistance) or eventually progress (secondary resistance), and there is accumulated evidence that in some patients ICIs can even favor tumor growth (hyperprogression) [53]. Because of this, different studies have been carried out to identify predictive factors for the efficacy of this type of treatment, as well as strategies to avoid resistance to it, with some of these studies showing that the composition of the intestinal microbiota modulates the activity, efficacy, and toxicity of ICIs.

#### 4.3.1. Anti-CTL-4

In patients treated with anti-CTLA4 antibodies, side effects are greater in those with a gut microbiota abundant in different *Firmicutes* species, such as *Faecalibacterium*, and a decreased abundance of *Bacterioides* [55,56]. In terms of treatment efficacy, in patients with metastatic melanoma, it was found that those whose gut microbiota was enriched in *Faecalibacterium* and other *Firmicutes* had longer progression-free survival and overall survival than those with microbiota rich in *Bacteroides* [55].

#### 4.3.2. Anti-PD-L1

The efficacy of the antibody targeting PD-L1 in the treatment of melanoma in mice is improved in the presence of a gut microbiota enriched in *Bifidobacterium* species. Additionally, oral administration to patients of a cocktail of bacteria of this species combined with the anti-PD-L1 antibody specifically increases the T-cell response and blocks melanoma growth, whereas, when the treatment is combined with antibiotics, the survival rate is lower [57].

#### 4.3.3. Anti-PD1

As was the case with anti-PD-L1 therapy, when combining anti-PD1 with antibiotics, the survival rate in patients is lower. In these patients, the responders to anti-PD1 treatment had a gut microbiota enriched in the *Akkermansia* and *Alistipes* genera [54]. Likewise, when analyzing the intestinal microbiota of patients with metastatic melanoma subjected to anti-PD-1 immunotherapy, a greater diversity and abundance of Faecalibacterium was observed in those with greater response to treatment and SSP, and a lower diversity and abundance of Bacteroilades in non-responders with lower SSP was observed [53].

## 5. Clinical Studies Dealing with Gut Microbiota and Breast Cancer

### 5.1. Completed Clinical Trials

More than half of the women who develop BC do not present any potential risk factors. In contrast, patients with a genetic predisposition or exposed to harmful environmental risk factors do not always develop this disease. Therefore, environmental factors must play a key role in the development of BC [58]. Indeed, factors such as diet, alcohol, and radiation have been associated with an increased incidence of BC [59].

The relationship between cancer and microbiota is not surprising, since altered host–gut microbiota interactions caused by dysbiosis seem to play an important role in carcinogenesis [60]. Many hypotheses suggest that the possible decrease in the metabolic ability of the microbiota and the weakness of the immune system are implied in the development of cancer [61]. Moreover, results from several studies show different profiles of the intestinal microbiota in BC patients compared to healthy controls. Such differences not only are related to the type and quantity of microbes that form the microbiota, but also to the activity of these microbes at the metabolic level, DNA damage, etc. [62].

The results of clinical studies dealing with the relation between gut microbiota and BC are summarized in Table 1. Regarding the methodology used in these trials, our understanding of the human microbiome has increased exponentially in the last decade, driven largely by advances in next-generation sequencing technologies and the application of metagenomic approaches [63]. Nowadays, two extensively used metagenome sequencing strategies are shotgun and PCR amplification of 16S rRNA gene and sequencing. BC microbiota has mainly been addressed by the latter [10,64,65,66,67,68,69,70,71,72,73], a strategy of gene sequencing that identifies and quantifies species or operational taxonomic units (OTUs).

In this respect, it is worthy to remark that therefore most of the studies reported in the last five years ignore the involvement of other microbial communities, such as fungi and viruses, despite the fact that these populations might also contribute to cancer development and aggressiveness. In addition, most of the clinical trials have been conducted with small sample sizes, and the accuracy of their conclusions remains to be confirmed.

BC patients usually exhibit a lower microbial diversity, as well as changes in the microbial composition. For instance, these women show increased levels of *Clostridiaceae*, *Faecalibacterium,* and *Ruminococcaceae*, as well as lower levels of *Dorea* and *Lachnospiraceae*, changes that may be explained by other risk factors such as adiposity and obesity [84].

In another clinical trial, Luu et al. described significant differences in the absolute numbers of total bacteria and of *Firmicutes*, *Faecalibacterium prausnitzii,* and *Blautia* in feces. These results correlated with the body mass index of women with early-stage BC, with a lower number of bacteria in overweight and obese patients [82]. Similarly, other authors have found that BC patients had a lower fecal relative abundance of *Firmicutes* and *Bacteroidetes* and a higher relative abundance of *Verrucomicrobla* and *Proteobacteria* [81]. Another study conducted by Frugé et al. in early-stage BC patients reported that body composition was inversely associated with *Akkermansia muciniphila*, and positively with interleukin-6 levels. These authors also reported that *Akkermansia muciniphila* relative abundance correlated with relevant health outcome parameters and were associated with favorable dietary changes [64].

Besides the microbial composition in fecal samples, the microbiota profile of breast tissue has also been studied. Differences in the relative abundance of various bacterial taxa and α-diversify have been observed in BC patients compared with healthy controls [66]. Moreover, in a study conducted in breast tissue samples from BC patients and healthy controls, *Propionibacterium* and *Staphylococcus* were depleted in tumors, showing negative associations with oncogenic immune features, while *Streptococcus* and *Propionibacterium* were positively correlated with T-cell activation-related genes [67]. Costantini et al. described *Proteobacteria*, *Firmicutes*, *Actinobacteria,* and *Bacteroidetes* associated with breast tumors, *Ralstonia* being the most prominent genus [76]. Meng et al., however, observed an increased representation of the genus *Propionicimonas* and the families *Micrococcaceae*, *Caulobacteraceae*, *Rhodobacteraceae*, *Nocardioidaceae,* and *Methylobacteriaceae* in malignant breast tumor tissues using a Chinese cohort of patients, although it is important to consider that these results are probably affected by the ethnic-specific characteristic of the studied population [75]. In contrast, Wang et al. reported no major changes in the overall diversity and microbiota composition of breast tissue when they compared patients with breast invasive carcinoma and healthy paired tissues [68]. However, these authors found differences in the microbiota of urine from BC patients, characterized by increased levels of Gram-positive organisms, including *Corynebacterium*, *Staphylococcus*, *Actinomyces*, and *Propionibacteriaceae*, and decreased abundance of *Lactobacillus.* Further, Nejman et al. found higher bacterial load and richness in breast tumor samples than those found in breast samples from healthy subjects [10]. In another study, Thompson et al. characterized the breast microbiota in neoplasm tissues and non-cancerous adjacent tissues from The Cancer Genome Atlas. Their results suggested a possible microbial compositional shift among the disease subtypes. The presence of *Proteobacteria* was increased in the tumor tissues, while *Actinobacteria* abundance increased in non-cancerous adjacent tissues. In addition, these authors found a possible association between *Listeria spp* and the expression profiles of genes involved with epithelial to mesenchymal transitions [69]. In this respect, the differences between tumor characteristics and stage have been addressed in recent years. For instance, no significant differences in α-diversity or phyla by estrogen/progesterone receptor status, tumor grade, stage, parity, and body mass index were found by Wu et al. However, these authors observed that particularly the HER2+ subtype (these cancers tend to grow and spread faster) women presented reduced α-diversity and *Firmicutes* abundance but increased abundance of *Bacteroidetes* in feces [79]. In addition, an increase in the abundance of the genus *Bosea* (phylum *Proteobacteria*) in tissue adjacent to breast tumor was associated with the tumor stage [70].

Not only bacteria, but other microbial populations, have been confirmed in the breast tumor tissue and/or tumor microenvironment. Some authors have suggested that many viral profiles could be associated with specific BC subtypes [19]. Banerjee et al. investigated the diversity of the microbiome in the four major types of BC (endocrine receptor-positive, triple positive, HER2+, and triple-negative BC). Two different patterns were detected, one for the triple-negative and triple-positive BC types, and another for the estrogen receptor-positive and HER2+ positive BC samples, compared to healthy breast control tissue [83]. Moreover, these authors detected the viral families, *Birnaviridae* and *Hepeviridae,* only in the triple-negative subtype, and *Nodaviridae* only in HER2+. Fungal signatures of *Filobasidiella*, *Mucor,* and *Trichophyton* were associated with estrogen or progesterone receptor-positive tumors. In contrast, healthy controls did not present *Ajellomyces*, *Alternaria*, *Cunninghamella*, *Epidermophyton*, *Filobasidiella*, *Rhizomucor,* and *Trichophyton*, while these fungi were detected in one or more BC types. These are remarkable findings, since hormones such as estrogens are responsible for the maintenance of homeostasis, reproduction, development, and/or behavior [61], and the collection of enteric microbial genes whose products are capable of metabolizing estrogens (the so-called estrobolome) is a mechanism specifically related to hormone-dependent cancers [60]. In fact, elevated endogenous estrogen levels have been shown to be associated with increased BC risk [85].

Associations between BC and estrogen levels might be correlated with differences in intestinal microbial composition among individuals [86]. Goedert et al. conducted a study in postmenopausal BC women (stages 0–1) and found that these patients showed decreased richness and α-diversity in their fecal microbiota with lower *Dorea* and *Lachnospiracea* and higher *Clostridiaceae* and *Faecalibacterium.* Of note, the richness was even lower in IgA-positive patients compared with IgA-negative patients. In the same study, these authors also found that estrogen levels in control age-matched postmenopausal women were directly correlated with their IgA-negative and microbiota α-diversity [71]. On the contrary, Jones et al. found no significant association between breast density and fecal microbiota in postmenopausal women (50–74 years old) with a normal mammogram. These authors also observed an inverse association between breast density and total urinary estrogens, and no association between mammographic density status and fecal microbiota β-diversity. These findings are of interest, since higher breast density has been associated with an increased risk of BC development [80]. In the trial NCT01461070 [87] (completed) conducted by the same group, the authors aimed to decipher the fecal microbiota and its association with systemic estrogens in postmenopausal women. This study also intended to determine differences in fecal microbiota profiles and urine estrogen levels between newly diagnosed postmenopausal BC cases and randomly sampled women. However, no data have been reported to date.

Zhu et al. found increased microbial gut diversity in BC patients compared with controls but no differences in relative abundance in gut microbiota between premenopausal BC patients and premenopausal controls. They also reported increased *Escherichia coli*, *Klebsiella* sp_1_1_55, *Prevotellaamnii*, *Enterococcus gallinarum*, *Actinomyces* ssp. HPA0247, *Shewanella putrefaciens* and *Erwinia amylovora*, as well as decreased *Eubacterium eligens* and *Lactobacillus vaginalis* in postmenopausal BC patients [74].

Microbial diversity differences depending on ethnicity in BC have also been reported in recent years [70,72]. Thyagarajan et al. described that when triple negative BC tumor tissue was compared to the matched normal tissue adjacent to the tumor, diversity (measured as Shannon index) was reduced in black non-Hispanic patients, while the white non-Hispanic cohort had an inverse pattern [70]. In addition, the phylum *Proteobacteria* was more abundant in normal tissue and tumor tissue in both white non-Hispanic and black non-Hispanic women; and *Firmicutes*, *Bacteroidetes* and *Actinobacteria* were less abundant. Overall both ethnicities presented an enrichment in family *Streptococcaceae* in triple negative BC [72].

One of the most important difficulties in the current treatment of BC is to counteract side effects and resistance to chemotherapy. Unfortunately, the mechanisms linking the response or resistance to chemotherapy are poorly understood and multifactorial, involving clinical and biological factors related to both the host and the tumor, and possibly to the patient’s psycho-social environment. Evidence points that chemotherapy causes devastating effects on microbial diversity and leads to dysbiosis and severe gastrointestinal toxicities. Moreover, some recent data associate alterations in the microbiome composition with the late effects of treatment in cancer survivors [88]. However, few studies have addressed the link between BC chemotherapy and its impact on gut microbiota. Chiba et al. found that women treated with neoadjuvant chemotherapy resulted in a significant increase in the abundance of *Pseudomonas* spp. as well as in a bacterial diversity reduction in breast tumor tissue. In the same study, a lower abundance of *Prevotella* in tumor tissue of non-treated patients was found [77]. In addition, Horigome et al. described an association of polyunsaturated fatty acids (PUFAs) with phyla *Actinobacteria* and *Bacteroidetes* in patients previously treated with chemotherapy, and an association of the genus *Bifidobacterium* with non-treated participants [78].

Finally, trial NCT03290651 [89] aims at improving the mammary microbial profile based on the supplementation with several strains of *Lactobacillus* in women at high risk of developing BC or survivors of this disease, although no results have been reported to date.

### 5.2. Ongoing Clinical Trials

As for currently ongoing trials, the database ClinicalTrials.gov (https://www.clinicaltrials.gov/ (accessed on16 July 2022), which details privately and publicly funded clinical studies conducted around the world, includes 13 registered clinical trials linking BC and microbiota in different stages of recruitment without no reported results to date (Table 2).

Concerning chemotherapy and microbiota, the observational clinical trial NCT04138979 aims to evaluate transcriptional changes in gut microbiota after cyclophosphamide chemotherapy during and after treatment [90]. Similarly, the pilot research trial NCT02370277 [91], also observational, evaluates the effects of chemotherapy (drug not indicated) on intestinal microbiota in newly diagnosed BC patients and associated with cancer recurrence. Finally, the clinical trial NCT03586297 [92], a prospective study, aims to determine the potential correlation between the pathologic responses in triple-negative BC patients treated with standard chemotherapy as well as the variability in the composition of intestinal and intra-tumoral microbiota.

In the clinical trial NCT02370277 [91], the authors aimed to determine the relationship between blood estrogen levels and gut microbiota. Importantly, the exposure to endocrine disruptors present in environmental contaminants might contribute to alter the microbiota and increase the risk of BC. This hypothesis is evaluated in the study NCT03885648 [93], being the first, to our knowledge, in which the contribution of bacteria, archaea, viruses, and fungi, together with the exposure to environmental contaminants, is addressed in BC patients.

As mentioned above, hormones also modulate behavior, a process that could be mediated by human microbiota. Thus, BC patients have a high risk of developing depression. Indeed, it is estimated that approximately 20%–45% of BC patients suffer depression after surgery. The of aim of trial NCT04303325 is to evaluate the postoperative depression, gut microbiota composition, and bi-spectral index [94] (a useful marker of anesthetic depth) data of patients undergoing BC surgery and treated with esketamine (a drug used as a general anesthetic and for treatment-resistant depression).

Of the active clinical trials annotated to date in ClinicalTrials.gov, two propose the use of probiotics, prebiotics, or synbiotics to modulate the microbiota in the setting of BC (NCT04139993 and NCT04784182 [95]). Trial NCT04139993 [96] evaluates the immunological effects of a novel oral Microbiome Restoration Therapy™ (MRT), RBX7455, in patients with stage I-III BC before undergoing surgery. In the same line, the trial NCT05113485 [97] outcomes are also to reduce sensitivity to C-reactive protein and to reduce visceral body fat and increase α- diversity of gut microbes by a highly-microbiota-accessible foods intake and aerobic exercise to reduce risk of BC recurrence. 

Similarly, in the trial NCT05000502, inactive BC survivors will perform aerobic exercise training, and an energy-balanced diet while changes in the gut microbiome will be assessed.

This is an interesting approach since, in general, cancer survivors experience more rapid declines in health-related quality of life, including physical and psychological comorbidities. In this regard, physical exercise is known to impact the composition and functional capacity of the microbiota with potential health benefits [100]. Thus, this might be an additional tool to re-balance the microbiota composition, as well as improve inflammation and immune parameters. This strategy could be applied not only in survivors, but also in patients under treatment. Moreover, a body of evidence has demonstrated that physical activity and exercise can prevent common mental disorders, such as depression and anxiety and promote mental health [101]. Hence, trials such as NCT04784182 [95] are of great interest since the anxiety symptoms that experience female BC survivors might be alleviated by daily consumption of a symbiotic supplement. Finally, another study has been designed with the same premise but without any nutritional supplementation: NCT04088708 [99], whose goal is to determine the effects of exercise on the gut microbiome in BC survivors and determine how these changes may relate to psychosocial symptoms such as fatigue.

## 6. Conclusions

Gut microbiota is closely related to cancer. Intestinal dysbiosis can favor certain microorganisms that inhabit our intestine to increase the risk of different types of cancer through numerous mechanisms such as virulence factors that degrade the products of tumor suppressor genes, toxins that damage DNA, generation of oxidative stress, activation of proinflammatory mechanisms, alteration of cell proliferation and survival pathways, and alteration of the immune system. When this microbiota is in equilibrium, a situation of eubiosis, it can act as a protective factor against cancer through the fermentation of dietary fiber and production of short-chain fatty acids, which allows maintaining the integrity of the intestinal mucosa and the immune system or triggering immune responses against tumor development. These pro- and anti-cancer mechanisms have been demonstrated in different types of cancer, not only in the intestine but also in other parts of the body, such as breast, liver, lung, or stomach cancer. In some of these cancers, in addition to these general mechanisms, there are some specific ones that can alter the development of the disease, such as the important role of the microbiota in the metabolism of estrogens, fundamental in BC, or the metabolism of bile acids, globally important in all tumors, but more specifically in liver cancer. In addition, numerous studies have shown the difference in the intestinal microbiota, both in quantity and diversity, in patients affected by any of these cancers compared to healthy individuals.

Different studies have proven the importance of the state of the intestinal microbiota in the activity and efficacy of anticancer therapies, to such an extent that different chemotherapy or immunotherapy drugs can be more or less effective when combined with antibiotics or probiotics. In addition to the efficacy of treatments, the state of the microbiota plays an important role in the susceptibility of the individual to side effects due to the toxicity of these therapies. Because of its influence on disease development and prognosis, the microbiota has become a target in the field of anticancer therapy. Although multiple clinical trials have been carried out or are ongoing to investigate the relationship between microbiota and BC from multiple perspectives, most of them are performed with small size populations and patients of different ages and tumor stages, which translates into controversial or noncomparable results. In our view, wider cohort studies and the use of standardized protocols to recognize possible microbial profiles that might be used as non-invasive biomarkers of BC are necessary. Additionally, the number of ongoing trials focused on physical exercise as a modulator of gut microbiota to improve quality of life and the immune system is remarkable. However, more studies that combine this approach with dietary patterns and are more focused on nutrition are necessary given the fact that obesity is deeply related to dysbiosis and cancer.

Finally, another important limitation to bear in mind regarding the clinical trials dealing with BC and microbiota is that none of them address the question of whether the changes in the microbiota that occur in this disease are a cause or an effect of the disease.

## Figures and Tables

**Figure 1 cancers-15-00443-f001:**
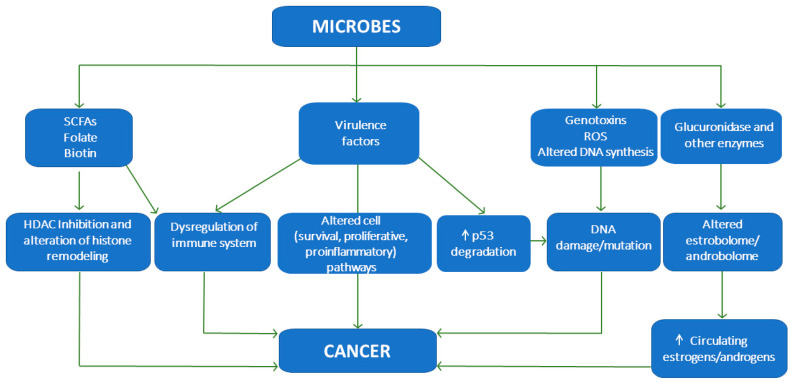
Several proposed mechanisms to understand the microbial influence on cancer. ↑ means increment.

**Figure 2 cancers-15-00443-f002:**
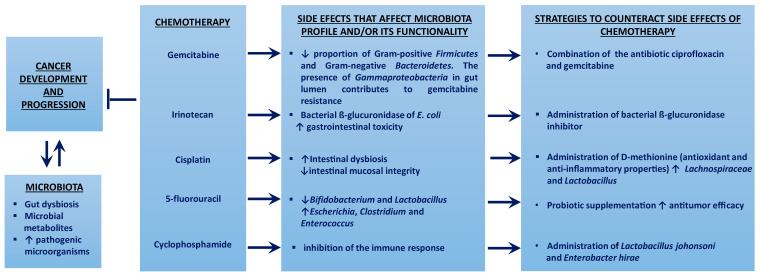
The gut microbiota affects cancer pathogenesis and the metabolism of chemotherapy drugs, conditioning both the response of cancer cells and the susceptibility of healthy cells. ↑ means increment; ↓ means decrease; ⱶ means inhibition.

**Table 1 cancers-15-00443-t001:** Clinical trials dealing with breast cancer and microbiota.

Study	Sampling Materials and Site	Microbiota Detection and OTU Picking Method	Sample Size	Main Findings
Nejman et al., 2020 [10]	Breast tumor samples from cancer patients. Breast samples from healthy subjects.	16S rRNA sequencing that amplifies five short regions along the 16S rRNA gene: the 5R 16S rRNA sequencing method. Greengenes database and Ribosomal Database Project classifier.	256 normal breast samples from healthy subjects. 355 breast cancer samples.	↑ Bacterial load and richness in breast tumor samples than those found in normal breast samples from healthy subjects. The microbiome of breast cancer is richer and more diverse than that of other tumor types.
Goedert et al., 2018 [71]	Fecal and urine samples.	16S rRNA gene amplicon sequencing: SILVA was used to assign sequences to OTU and HPLC /MS used to assign 16S rRNA gene sequences to OTUs.	48 postmenopausal breast cancer women (75% stage 0–1, 88% estrogen-receptor positive). 48 contemporaneous women, postmenopausal, normal-mammogram.	Women with breast cancer had non-significantly elevated estrogen levels. Estrogens in healthy control (but not cases) subjects were directly correlated with their IgA-negative microbiota α-diversity. Prostaglandin E metabolite levels were not associated with tumor status, estrogen levels, or α-diversity. breast cancer patients. ↓ α-diversity and altered composition of both their IgA-positive and IgA-negative fecal microbiota in breast cancer. ↑ Microbial IgA-positive imputed Immune System Diseases metabolic pathway genes. Cases women: ↑ Levels of *Clostridiaceae*, *Faecalibacterium*, and *Ruminococcaceae.* ↓ Levels of *Dorea* and *Lachnospiraceae.*
Frugé et al., 2020 [64]	Serum and fecal samples.	16S-V4 rRNA gene amplicon sequencing. RDP classifier.	32 female breast cancer patients randomized to weight-loss or attention-control arms from the time of diagnosis to tumorectomy.	In the early stage of breast cancer, body composition is associated with *Akkermansia muciniphila*, microbiota diversity, and interleukin-6 level. Different composition and functions of the gut microbial community between postmenopausal breast cancer patients and healthy controls. *Akkermansia muciniphila* is related to relevant health outcome parameters and to favorable dietary changes.
Zhu et al., 2018 [74]	Fecal samples.	Shotgun metagenomic analysis.	Premenopausal women: 18 breast cancer patients. 25 healthy controls. Postmenopausal women: 44 breast cancer patients. 46 healthy controls.	↑ Microbial diversity in breast cancer patients than in controls. No differences in relative abundance in gut microbiota between premenopausal breast cancer patients and premenopausal controls. In postmenopausal breast cancer patients: ↑ *Escherichia coli*, *Klebsiella sp_1_1_55*, *Prevotellaamnii*, *Enterococcus gallinarum*, *Actinomycessp.* HPA0247, Shewanella putrefaciens, and Erwinia amylovora, and ↓ *Eubacterium eligens* and *Lactobacillus vaginalis.*
Klann et al., 2020 [66]	Breast tumors from cancer patients. Breast samples from healthy subjects.	16S rRNA V1–V2 hypervariable regions. RDP classifier and verified against the Greengenes database.	Bilateral normal breast tissue samples (n = 36) collected from 10 women who received routine reduction mammoplasty. Archived breast tumor samples (n = 10) obtained from a biorepository.	Breast cancer samples differed in microbiota composition across individual women. The most abundant phyla in both tumor and normal tissues were *Bacteroidetes*, *Firmicutes*, *Proteobacteria*, and *Actinobacteria.* Differences in the relative abundance of various bacterial taxa between groups. ↑ α-diversity in normal compared to tumor samples.
Meng, et al., 2018 [75]	Breast tissue samples.	V1-V2 16S rRNA Sequencing.	22 Chinese patients with benign tumor and 72 malignant breast cancer patients.	Levels of *Propionicimonas*, *Micrococcaceae*, *Caulobacteraceae*, *Rhodobacteraceae*, *Nocardioidaceae,* and *Methylobacteriaceae*, in breast cancer tissues. *Bacteroidaceae* and *Agrococcus* associated with malignancy.
Costantini, L. et al., 2018 [76]	Breast tissue samples.	V3 16S-rRNA gene amplicons Sequencing.	16 Mediterranean patients with breast cancer (12 samples were collected from core needle biopsies (CNB) and seven from surgical excision biopsies (SEB); three patients were processed with both procedures). Fresh tumor breast tissue and paired breast healthy tissue.	*Ralstonia* was the most prominent genus in tumor breast tissue. No differences between healthy adjacent breast tissue and breast cancer tissue.
Chiba, A. et al., 2019 [77]	Snap-frozen breast tumor tissue.	V4 16S rRNA amplicon sequencing (Illumina Miseq). Pipelinee: Mothur (v.1.39.5) Microarray for confirmation.	An amount of 15 women with breast cancer who were treated with neoadjuvant chemotherapy, 18 women with no prior therapy at the time of surgery, and nine women who had tumor recurrence.	Presence of *Pseudomonas* spp. in breast cancer tissue after neoadjuvant chemotherapy. Presence of *Prevotella* in the tumor tissue from non-treated patients. Presence of *Brevundimonas* and *Staphylococcus* in the primary breast tumors in patients developing distant metastases.
Horigome, A. et al., 2019 [78]	Capillary blood and fecal samples.	V3-V4 region of the bacterial 16S rRNA gene sequencing. (Illumina Miseq). Pipeline: QIIME2 Gas chromatography for Fatty acid composition.	124 participants (46% history. of chemotherapy). (123 women and one man).	*Actinobacteria* and *Bacteroidetes* were associated with PUFAs in patients previously treated with chemotherapy. *Bifidobacterium* was associated to PUFAs in participants with no history of chemotherapy
Wu et al., 2020 [79]	Breast tissue and fecal samples (collected prior to chemotherapy).	16S rRNA gene amplification and sequencing of the V3 and V4 hypervariable regions.	37 breast cancer patients.	No differences in α-diversity or phyla differences by estrogen/progesterone receptor status, tumor grade, stage, parity, and body mass index. HER2+ women showed: ↓ α-diversity, *Firmicutes* abundance ↑ Abundance of *Bacteroidetes.* Early menarche associated with: ↓ OTU. ↓ Abundance of *Firmicutes.* ↑ High total body fat.
Jones et al., 2019 [80]	Urine and fecal samples.	16SrRNAgeneV3-V4hypervariable region. OTUs were assigned by Ribosomal Data Project Naïve Bayesian classifier.	54 postmenopausal women. (50–74 years old) with normal mammogram.	No association between breast density and fecal microbiota. Total urinary estrogens were strongly and inversely associated with breast density. Fecal microbiota α-diversity and richness did not differ between women with high *versus* low mammographic density.
Yoon et al., 2019 [65]	Fecal samples.	16S rRNA gene V3-V4 region. Greengenes database.	121 female participants between the ages of 32 and 78 who underwent a positron emission tomography PET/CT scan.	The physiologic intestinal uptake was positively correlated with the relative abundance of the genus *Citrobacter*, while negatively correlated with the unclassified *Ruminococcaceae.*
Ma et al., 2020 [81]	Fecal and blood samples.	16S rDNA amplicon sequencing Mothur method and the SSUrRNA database of SILVA.	25 breast cancer patients. 25 patients with benign breast disease.	In breast cancer group: ↓ Relative abundance of *Firmicutes* and *Bacteroidetes.* ↑ Relative abundance of *verrucomicrobla*, *Proteobacteria* and *Actinobacteria*↓ *Faecalibacterium*, which was negatively correlated with various phosphorylcholines.
Tzeng et al., 2021 [67]	Breast tissue samples.	Bacterial 16S rRNA gene V3–V4 and V7–V9 regions. Amplicon sequence variants (ASVs) were then classified against SILVA.	221 patients with breast cancer and 87 patients without breast cancer.	*Anaerococcus*, *Caulobacter*, and *Streptococcus,* predominant in benign tissue networks, were absent from cancer-associated tissue. *Propionibacterium* and *Staphylococcus* were depleted in tumors and showed negative associations with oncogenic immune features. *Streptococcus* and *Propionibacterium* correlated positively with T-cell activation-related genes. *Pseudomonas* constituted a wide proportion of the breast microbiome in tumor vs. other tissues, and *Proteus* was the second most abundant genus in tumor tissue but absent from non-tumor tissues.
Thyagarajan et al., 2020 [70]	Breast cancer and matched normal tissue adjacent to tumor samples.	16S rRNA gene-based sequencing. SILVA 16S rRNA database.	Six White non-Hispanic (WNH) of which two were tumor and two normal adjacent tissue. Seven Black non-Hispanic (BNH), triple-negative breast cancer (TNBC) Seven WNH, TNBC. Three BNH and triple-positive breast cancer (TPBC).	Microbial diversity was significantly lower in BNH TNBC tumor tissue as compared to matched normal tissue adjacent to the tumor zone. WNH cohort had an inverse pattern for the Shannon index, when TNBC tumor tissue was compared to the matched d normal tissue adjacent to the tumor. Unweighted PCoA revealed distinct clustering of tumor and d normal tissue adjacent to tumor microbiota in both BNH and WNH cohorts.
Smith et al., 2019 [72]	Breast tissue.	16S rRNA gene sequencing. Greengenes as the reference database.	An amount of 83 breast tissue samples, of which pathologically adjacent normal breast tissues (normal pair) were obtained from 11 breast cancer patients. 64 breast tissue samples from women with stages I-IV breast cancer and eight from healthy women who underwent breast reduction mammoplasty. Approximately 24% of the study participants were NHB, 75% NHW, and 64% were premenopausal.	*Proteobacteria* was most abundant in normal tissue adjacent to tumor and breast tumors from NHB and NHW women with fewer *Firmicutes*, *Bacteroidetes*, and *Actinobacteria.* ↑ Abundance of genus *Ralstonia* in NHB women compared to NHW tumors. Enrichment of family *Streptococcaceae* in TNBC. ↑ Abundance of genus *Bosea* (phylum Proteobacteria) associated with the tumor stage.
Luu et al., 2017 [82]	Feces from women with early-stage breast cancer.	qRT-PCR.	31 women with breast cancer [ER/PgR+ (90%), HER2+ (15%)].	In the fecal samples, *Firmicutes* and *Bacteroidetes* were the most abundant phyla. ↑Richness of *Bacteroidetes*, *Clostridium coccoides cluster*, *C. leptum cluster*, *F. prausnitzii*, and *Blautia* spp. In clinical stage groups II/III compared with clinical stages 0/I *Blautia* spp. was associated with more severe histoprognostic grades. ↓ Total bacteria and three groups: *Firmicutes*, *Faecalibacterium prausnitzii* and *Blautia* spp. in overweight and obese women.
Wang et al., 2017 [68]	Urine and bilateral breast tissue from each control patient, and tumor and ipsilateral adjacent normal breast tissue for cases.	Illumina 16S V3-V4 rRNA amplification. OTUs were assigned using Greengenes database, specific method not disclosed.	An amount of 50 patients and 20 healthy controls.	No significant difference in overall diversity in microbiota content (number of observed OTUs) was detected in breast tissue from cancer and control women. ↓ Relative richness of *Methylobacterium* was found in women with breast cancer. Differences in the urinary microbiota of women with breast cancer: ↑ Abundance of *Corynebacterium*, *Staphylococcus*, *Actinomyces*, and *Propionibacteriaceae* gram-positive bacteria. ↓ Abundance of genus *Lactobacillus.*
Thompson et al., 2017 [69]	Breast tumor tissues and normal adjacent tissues from The Cancer Genome Atlas.	16S-V3-V5 rRNA amplified, metagenome Seq package. Greengenes database.	An amount of 668 tumor tissues (HER2+, ER+ and TNCBC) and 72 normal adjacent tissues.	The most abundant phyla in breast tissues were *Proteobacteria*, *Actinobacteria*, and *Firmicutes.* In tumor samples, the most predominant phyla were *Proteobacteria* and *Actinobacteria* in normal tissue. *Mycobacterium fortuitum* and *Mycobacterium phlei* were two of the prevalent species observed differentially abundant in the *tumor samples.* ↑Prevalence of *Escherichia coli* in the breast tissues.
Banerjee et al., 2018 [83]	Breast cancer tissues (cases), breast control tissues from healthy individuals (reduction surgeries).	PathoChips array.	Breast cancer [ER+ (n = 50), HER2+ (n = 34), triple positive (n = 24), TNBC (n = 40)], and normal breast tissue (n = 20).	Unique viral, bacterial, fungal and parasitic signatures were found for each of the breast cancer types. The triple-negative and positive samples showed distinct microbial signature patterns than the ER and HER2 positive breast cancer samples. The most prevalent bacterial signatures were *Proteobacteria* followed by *Firmicutes*. The *Mobiluncus* family was detected in all four types.

Abbreviations: BNH, black non-Hispanic; CNB, core needle biopsies; ER+, estrogen receptor-positive; HER2+, human epidermal growth factor receptor 2 positive; n; number; OUT, operational taxonomic unit; PCoA, principal coordinate analysis; PgR+, progesterone receptor-positive; PUFAs, polyunsaturated fatty acid; RDP, Ribosomal Database Project; SEB, surgical excision biopsies; TNBC, triple-negative breast cancer; TPBC, triple-positive breast cancer; WNH, white non-Hispanic. ↑ means increased and ↓ means decreased.

**Table 2 cancers-15-00443-t002:** Ongoing clinical studies registered on ClinicalTrials.gov dealing with breast cancer and microbiota.

Title	NCT Number and Location	Study Type	Objective	Type of Sample and Microbiota Detection Method	Time Frame	Intervention	Sample Size and Participants´Characteristics	Status of the Recruitment
Effects of Chemotherapy on Intestinal Bacteria in Patients With Newly Diagnosed Breast Cancer [91].	NCT02370277 United States of America.	Observational Case-Control.	To establish changes in gut microbiota related to chemotherapy treatment.	Stool samples, measurement of the number of taxonomic groups.	Baseline to four months after final adjuvant (or neoadjuvant) chemotherapy course.	N/A.	n = 36 ≥18 years Gender: Female Current patients of breast cancer.	Completed
Intestine Bacteria and Breast Cancer Risk [87].	NCT01461070 United States of America.	Observational Case-Control.	To establish association between fecal microbiome and systemic estrogens levels.	Stool samples, 16S rRNA metagenomics sequencing.	Cross-sectional	N/A	n = 175 50–69 years Gender: Female Current patients of breast cancer and healthy women.	Completed
Persistent Post Surgical Pain in Women With breast cancer [98].	NCT02266082 United States of America.	Observational Cohort.	To establish changes in gut microbiome potentially associated with pain after surgery and mental disorders.	Stool samples, 16S rRNA, metagenomics sequencing.	Baseline, three months and six months post-surgery.	N/A.	n = 5 40–75 years Gender: Female.	Completed
Effect of Esketamine on Postoperative Depression Gut Microbiota Bispectral Index Data of Depression Patients Undergoing Breast Cancer Operation [94].	NCT04303325 China.	Interventional Randomized.	To detemine the effect of esketamine on gut microbiome potentialy associated with depresion mental disorders.	Stool samples, method not indicated.	Baseline to 72 h post-surgery.	Esketamine or saline solution.	n = 36 18–65 years Gender: Female Current patients of breast cancer.	Not yet recruiting.
Anti-anxiety Biotics for Breast Cancer Survivors [95].	NCT04784182 United States of America.	Interventional Randomized.	To evaluate the effect of probiotics in anxiety symptoms in breast cancer survivors.	Stool samples, 16S rRNA metagenomics sequencing.	Four weeks.	Daily dietary supplementation with at least five billion CFU per day of total bacteria including *Lactobacillus helveticus* and *Bifidobacterium longum* and prebiotic containing 4 g of fructooligosaccharides for four weeks.	n = 48 50–75 years Gender: Female breast cancer survivors.	Not yet recruiting
Intestinal Microbiota of Breast Cancer Patients Undergoing Chemotherapy [90].	NCT04138979 China.	Observational Case-Control.	To determine transcriptional changes in gut microbiota during and after chemotherapy.	Stool samples, 16S rRNA metagenomics sequencing.	Baseline, 1, 7, 14, 22, 29, 36, 44, 51, 58, 66, 73, and 80 days.	Cyclophosphamide.	n = 80 18–65 years Gender: Female Current patients of breast cancer.	Recruiting
Breast Cancer and Its Relationship With the Microbiota [93].	NCT03885648 Spain.	Observational Case-Control.	To detemine the contribution of bacteria, archaea, viruses and fungi together with exposure to environmental contaminants to the risk of breast cancer.	Stool and mamary samples, shotgun metagenomics sequencing.	Post surgery	N/A.	n = 200 25–70 years Gender: Female Current patients of breast cancer.	Recruiting
A Pilot Trial of Preoperative Oral Microbiota-based Investigational New Drugs [96].	NCT04139993 United States.	Interventional Non Randomized.	Evaluation of RBX7455 effect on the restoration of microbiota.	Stool samples, bacterial taxonomy identification.	An amount of ≤2 days prior to surgery, as well as eight weeks and six months after treatment.	RBX7455, a novel microbiome restoration therapy™.	n = 30 ≥18 years Gender: Female Current patients of breast cancer.	Recruiting
Gut and Intratumoral Microbiome Effect on the Neoadjuvant Chemotherapy- induced Immunosurveillance in Triple Negative Breast Cancer [92].	NCT03586297 United States of America.	Observational Cohort.	To determine correlations between responses in triple-negative breast cancer patients to standard chemotherapy and the variations in microbiota profile.	Stool and tumoral samples, 16S rRNA and shotgun metagenomics sequencing.	Completion of chemotherapy, (approximately 18 weeks).	N/A.	n = 49 ≥18 years Gender: Female Current patients of breast cancer.	Recruiting
Probiotics and Breast Health Study [89].	NCT03290651 Canada.	Interventional Randomized.	To test the hypothesis that probiotics can reach the breast tissue and contribute to displace the harmful bacteria.	Mamary samples, method not indicated.	An amount of 90 days post collection.	Dietary supplement: One capsule containing 2.5 billion CFU of *Lactobacillus rhamnosus* GR-1 and *Lactobacillus reuteri* RC-14 or placebo for 90 days.	n = 40 Children, adult an older adult Gender: Female Women at high risk of developing breast cancer who have never had breast cancer.	Recruiting
Gut Microbe Composition, Exercise, and Breast [99].	NCT04088708 United States of America.	Interventional Randomized.	To determine the effects of exercise on the gut microbiome in breast cancer survivors and how these changes may relate to psychosocial symptoms such as fatigue.	Stool sample, method not indicated.	Baseline, 5, 10, 15 weeks.	Aerobic exercise training.	n = 126 18–70 years Gender: Female breast cancer survivors.	Recruiting
A Fiber-diverse, Anti-inflammatory Diet and Aerobic Exercise Reduce Risk of Breast Cancer Recurrence [97].	NCT05113485 United States of America.	Interventional Randomized Intervention	To conduct two parallel, three-month behavior change interventions, contrasting the six-count highly-microbiota-accessible foods approach with the traditional diabetes prevention program calorie restriction approach To design a ramped-up randomized factorial trial.	Stool sample, method not indicated.	Baseline and six-month follow-up assessments of: low grade systemic inflammation, body composition including visceral fat estimation, cardiorespiratory fitness, inflammatory and cardiometabolic biomarkers.	Highly-Microbiota-Accessible Foods approach with the traditional or Diabetes Prevention Program calorie restriction	n = 30 50–75 years Gender: Female breast cancer survivors. interested in losing excess body fat.	Recruiting

Abbreviations: CFU, colony forming units; HPLC/MS, high-performance liquid chromatography/mass spectrometry; N/A, non-applicable; n, number.

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
