# Peer review of "Gut Microbiota and Breast Cancer: The Dual Role of Microbes"

_cancers, 2023, doi:10.3390/cancers15020443_

Round 1

Reviewer 1 Report

Thank you for submitting this comprehensive review.  However; I would suggest to add a section to discuss in details the  effect of microbiota on the clinical outcome, chemotherapy resistance 

Author Response

Thank you very much for your comments. As suggested, we have added a new section discussing the relationship between microbiota and chemotherapy and immunotherapy (section 4). This section includes a new figure (figure 2).

Changes are highlighted in green.

Reviewer 2 Report

The manuscript by Ana I Álvarez-Mercado et al. “Gut microbiota and breast cancer: The dual role of microbes” summarized the mechanisms by which the gut microbiota may cause cancer in general, and breast cancer in particular, and to compile clinical trials that address alterations or changes in the microbiota of women with breast cancer. Although the review is well-written, there are some limitations as mentioned below: 

1. The authors should add more information in “Conclusions”.

2. The authors should add more figures and tables.

3. The grammar and word errors need to be corrected thoroughly in the manuscript.

Author Response

Thank you very much for your comments. Below is a point-by-point letter answering your concerns. Changes are highlighted in green.

  1. The authors should add more information in “Conclusions”. As suggested we have extended this section.
  2. The authors should add more figures and tables. A new section discussing the relationship between microbiota and chemotherapy and immunotherapy (section 4) has been added to the manuscript. This section includes a new figure (figure 2).
  3. The grammar and word errors need to be corrected thoroughly in the manuscript. We have thoroughly revised the manuscript and corrected the errors.

Round 2

Reviewer 1 Report

Thank you for your re-submitting the revised manuscript which adequately addressed my previous comments.

Reviewer 2 Report

Accept.